# Violence as the Most Frequent Cause of Oral and Maxillofacial Injuries among the Patients from Low- and Middle-Income Countries—A Retrospective Study at a Level I Trauma University Emergency Department in Switzerland

**DOI:** 10.3390/ijerph17134906

**Published:** 2020-07-07

**Authors:** Triantafillos Loutroukis, Ekaterini Loutrouki, Jolanta Klukowska-Rötzler, Sabine Koba, Fabian Schlittler, Benoit Schaller, Aristomenis K. Exadaktylos, Michael Doulberis, David S. Srivastava, Silvana Papoutsi, John Patrik M. Burkhard

**Affiliations:** 1Department of Emergency Medicine, Inselspital, University Hospital Bern, University of Bern, CH-3010 Bern, Switzerland; loutroukis@hotmail.com (T.L.); eloutrouki@hotmail.com (E.L.); Jolanta.klukowska-roetzler@insel.ch (J.K.-R.); Aristomenis.Exadaktylos@insel.ch (A.K.E.); michael.doulberis@ksa.ch (M.D.); DavidShiva.Srivastava@insel.ch (D.S.S.); 2Department of Cranio-Maxillofacial Surgery, Inselspital, University Hospital Bern, University of Bern, CH-3010 Bern, Switzerland; koba.sabine@icloud.com (S.K.); fabian.schlittler@insel.ch (F.S.); benoit.schaller@insel.ch (B.S.); 3Division of Gastroenterology and Hepatology, Medical University Department, Kantonsspital Aarau, CH-5001 Aarau, Switzerland; 4Department of Visceral Surgery and Medicine, Inselspital, University Hospital Bern, University of Bern, CH-3010 Bern, Switzerland; silvana.ingold@gmail.com

**Keywords:** immigrants, low-income, middle income, facial trauma, maxillofacial trauma, oral health, dental trauma

## Abstract

Preventive strategies can be developed by gathering more information about oral and maxillofacial injuries and oral pathologies in immigrants from low- to middle-income countries (LMIC). Additional information on the quality of care can also improve the allocation of clinical resources for the management of these patients. We studied immigrants from LMIC who presented in the emergency department (ED) at Berne University Hospital with dental problems or oral or maxillofacial injuries. The patient data included age, gender, nationality, the etiology and type of trauma and infection in the oral-maxillofacial area, and overall costs. The greatest incidence of maxillofacial injuries was observed in the age group of 16–35 years (*n* = 128, 63.6%, *p* = 0.009), with males outnumbering females in all age groups. Trauma cases were most frequent in the late evening and were mostly associated with violence (*n* = 82, 55.4%, *p* = 0.001). The most common fracture was fracture of the nose (*n* = 31). The mean costs were approximately the same for men (mean = 2466.02 Swiss francs) and women (mean = 2117.95 Swiss francs) with maxillofacial injuries but were greater than for isolated dental problems. In conclusion, the etiology of dental and maxillofacial injuries in immigrants in Switzerland requires better support in the prevention of violence and continued promotion of oral health education.

## 1. Introduction

Migration is a social determinant of health, particularly oral health [1]. In recent years, a steadily growing number of immigrants and refugees from low-, low-middle-, and upper-middle-income countries (LMIC) have moved to Europe as a consequence of political instability, socioeconomic disparities, and environmental events [2]. The mean value for the proportion of migrants is 8.6% in the overall European population, but the maximum value is in Switzerland with 23.5%. According to the Federal Statistical Office, there were 2,165,000 immigrants in Switzerland in 2018, corresponding to 30.2% of the population [3]. Inevitably, this has an impact on local health policy, as the different cultural, environmental, lifestyle, genetic, and linguistic background of patients from LMIC influence the sociodemographic structures, including hospital Emergency Departments (ED) [4]. 

Only a small proportion of immigrants possess health insurance and often struggle to access medical care in their new countries. This is associated with regional differences in policy implementation and the high cost of insurance premiums [5]. In Switzerland, private health insurance is obligatory for access to healthcare—with the exception of dental treatment [5,6]. 

The higher incidence of dental disease and lower use of dental care among immigrants, compared to the local population, is a serious public health matter in many other countries [7]. It is not entirely clear whether this has an impact on the Swiss health care system, as the costs of dental treatment are generally paid by the patient. In Switzerland, dental emergencies are usually not managed by hospitals EDs or maxillofacial surgeons but are covered by general practice (GP) clinics, the emergency dental service, and the dental clinics of the University hospitals. In this context, Bern has defined its own system for dental emergencies, depicted in Figure 1. This is intended to relieve routine clinical work by preventing unnecessary visits, as well as avoiding the waste of medical resources and additional costs. It is essential that immigrant populations be educated in the use of dental care and oral health habits, with strong social support and a large community network [8,9,10].

In contrast, traumatic injury to the dental and maxillofacial structures following blunt trauma is a common reason for presentation to hospital EDs and is fully covered by insurance. This type of injury is often associated with concomitant injuries and may cause permanent functional and aesthetic deficits—with a dramatic impact on the patient’s quality of life [11,12]. The cultural differences, the high socioeconomic standard, lifestyle, and the alpine surroundings may influence the etiology of facial trauma in Switzerland [13]. There have only been a few studies on dental and maxillofacial injuries in the Swiss population, and none on immigrants to Switzerland [13,14,15,16,17].

Epidemiological data are of central importance and provide a fundamental basis for the evaluation of access to treatment, resource allocation, and planning within the health services. They are essential for the development of preventive strategies and the provision of information about the quality of care [18]. Violence is considered to be frequent among people of lower socioeconomic status. However, only a few relevant studies have been published [19]. It has been proposed that violence is common among patients from low- and middle-income countries. Data on immigrants from LMIC to Switzerland are sparse. The purpose of this study was to identify and categorize oral health issues and maxillofacial injuries in immigrants from low- and middle-income countries, as well as the costs incurred. 

## 2. Materials and Methods 

### 2.1. Study Design

This single-center retrospective cohort study was based on the demographic- and health-related data of 201 patients from LMIC who received care for dental problems and/or craniomaxillofacial injuries at the Level I interdisciplinary university ED at the Inselspital Bern. Patients younger than 16 years of age are always treated by the pediatric hospital and were not included in this study.

### 2.2. Data Collection and Extraction

All data were extracted from medical records and files stored in the clinical database system (E.care ED, ED 2.1.3.0, E.care BVBA, Turnhout, Belgium) from January 2016 until September 2019. Relevant information included age, gender, diagnosis, localization, nationality groups, time of admission, referral status, hospital stay, recording division, etiology and type of trauma or infection, and overall costs were also obtained. No distinction was made between the disciplines from which the medical records were written. Patients who were not examined by a physician were excluded from this study.

All patients were classified into the following categories: Age (16–25, 26–35, 36–45, 46–55, 56–65, 66–75, 76–85), gender (male, female), nationality groups (the classification of the countries was based on income according to World Bank Country and Lending Groups: Low-income countries ($1025 or less), low-middle-income countries ($1026 to $3995), upper-middle-income countries ($3996 to $12,375)), time of admission (year, weekday/weekend, daytime), triage level (1, 2, 3, 4, 5), and type of referral (self-referral, ambulance, Swiss air rescue, general practitioner, police/prison, external hospital, and internal referral). Furthermore, the costs were divided into two subgroups in order to distinguish the total costs from the maxillofacial or dental costs. 

### 2.3. Classification: Primary, Secondary and Tertiary Dentistry 

The classification of dental problems was carried out according to the Bern model, as illustrated in Figure 1. A distinction was made between primary, secondary, and tertiary dentistry, whereby primary and secondary dentistry are, by definition, covered by the family dentist or the dentist with a subspecialty training.

Tertiary dental emergencies include the care of dental patients who require in-patient treatment such as procedures under general anesthesia in a well-equipped operating theatre, which are covered by maxillofacial surgeons or specifically need monitoring in an intensive care unit, or have to be treated by other specialist disciplines due to their general condition, recognized previous diseases or concomitant injuries.

### 2.4. Triage System

At the ED, patients are routinely triaged using an abbreviated version of the Manchester Triage System, named the Swiss Emergency Triage Scale [20,21]. This triage system classifies the urgency of a treatment for patients in five different priority levels: 1—Acute life threating problem (immediate treatment required), 2—high urgency, 3—urgency, 4—less urgency, 5—nonurgency. Specially trained nurses work on the basis of a defined algorithm for classification, pursuant to the patient’s reported complaints and the treatment priority—with the aid of fixed rules that take into account the vital signs. 

### 2.5. Statistical Analsysis

Descriptive statistics were utilized to determine the frequencies and percentages for dichotomous variables, the mean values, standard deviation and ranges of numerical variables. Categorical variables were analyzed using the chi-square (χ^2^) test and Fisher’s exact test. The relationship between two variables was evaluated using the Bravais-Pearson correlation. Significance was set at *p* < 0.05 (two-tailed). Statistical analysis was performed using the Statistical Package for Social Sciences for Macintosh (SPSS^®^ Version 24.0, IBM Corp., Armonk, NY, USA).

### 2.6. Ethical Considerations

The Cantonal Ethics Committee of Bern approved this study (reference number: BE 2018-00198) and followed the guidelines of the Declaration of Helsinki and ethical principles for conducting medical research with human subjects [22]. Data were handled according to the standards of the Ethics Committee and Swiss law. No individual informed consent was obtained. The analysis was carried out with anonymized data.

## 3. Results

Over a period from January 2016 until September 2019, 201 patients with migration background were admitted to the ED with oral or craniomaxillofacial (CMF) problems. All patients were divided into two categories: Dental patients and CMF patients. These categories represent the allocation of cases by the discipline, i.e., cases in the dental category were attended or referred to dentists. Cases in the CMF category were managed by maxillofacial surgeons and/or other specialists for the head and neck or other areas.

### 3.1. Age, Gender, and Nationality Distribution 

Patient’s ages ranged from 16 to 81 years, with a mean age of 33.67 (standard deviation, SD = 12.76). More than half of the patients were younger than 35 years (*n* = 128, 63.6%). Elderly patients over 65 years of age accounted for 1.5% of the total number. More than half of the patients were male (*n* = 139, 69.2%) and 30.8% female (*n* = 62). The most frequent nationalities were Sri Lankan (*n* = 23, 11.4%), followed by Syrian (*n* = 16, 8%), Moroccan (*n* = 15, 6.5%), North Macedonian, Albanian, and Iraqi (*n* = 13, 6.5%). Only 13.9% (*n* = 28) of the patients were from low-income countries. Patients from countries with low-middle- or upper-middle-incomes were more frequent. There were similar numbers of patients in the two categories. Patients from countries with low-middle- or upper-middle-income made up the largest group of dental and CMF patients. The data are summarized in Table 1 and Figure 2.

### 3.2. Type of Referral

In all categories, more than half of the patients were self-referring (*n* = 128, 63.7%), corresponding to 86.8% (*n* = 46) of all dental and 55.4% (*n* = 82) of all CMF visits, and 42 patients (20.9%) were referred by ambulance. This was therefore the second largest referral type and was mainly made up of CMF patients. Other reported referrals were by external physicians/private practitioners, external/internal hospitals, the police, or Swiss air ambulance, as displayed in Table 2.

### 3.3. Manchester Triage System (MTS) & the Bern Model

Most patients were triaged as patients with serious, but apparently stable conditions (MTS: 3), corresponding to 79.2% of dental patients, 85.8% of maxillofacial patients, and 84.1% combined patients. No cases with triage level 1 or 5 were described. A summary is given in Table 2.

According to the definition of the Bern Model, from a total of 78 patients with dental problems, 53 patients (67.9%) could have been treated directly by a dentist (primary/secondary dentistry) and did not have to present in an ED. Most of the patients (86.6%) referred themselves to the ED. Eight (8) patients were in the category of tertiary dentistry (3.4%).

### 3.4. Time and Reason for Admission and Trauma Mechanism

The annual distribution revealed an approximately constant number of patient presentations in the ED (*p* = 0.892). Presentations were slightly more frequent in the afternoon and evening (*p* = 0.947) without a significant difference between weekends and weekdays (*p* = 0.354). Self-referral was significant in patients with poor oral health (*r* = 285, *p* = 0.047). The most frequent reason for presentation in the ED for dental problems was carious teeth (*n* = 44, 73.1%, *p* = 0.001). Isolated dental emergencies accounted for 26.4% (*n* = 53), with more than half being referred for tooth pain (*n* = 34, 64.2%). Eleven (11) patients had an infection with classic signs of inflammation and/or manifest abscess, whereas six of these (4.1%) had to be hospitalized and operated under general anesthesia. Dental conditions are summarized in Figure 3. 

Blunt dental trauma occurred in 23 cases. Five patients presented with isolated tooth fractures and three patients with luxation injuries. This most often occurred in the context of a physical dispute (*n* = 4, 7.5%). The remaining patients suffered combined dental and maxillofacial injuries.

A total of 137 patients suffered injuries to the maxillofacial area, which accounted for 92.6% of all visits. Interpersonal violence was the major cause by far for maxillofacial injuries (*n* = 86, 59.3%, *p* = 0.001), with significantly more facial injuries (soft tissue laceration and facial fracture) than tooth fractures. Ten (10) cases (6.9%, *p* = 0.034) of domestic violence against women were registered. The propensity to violence was much higher among men than women. Less frequent reasons for CMF injuries were falls from standing (*n* = 20, 13.8%, *p* = 0.010), road traffic injuries (*n* = 7, 4.8%, *p* = 0.079), work related injuries (*n* = 8, 5.5%, *p* = 0.274), sport injuries (*n* = 11, 7.6%, *p* = 0.020), bicycle accidents (*n* = 6, 4.1%, *p* = 0.105), falling down stairs (*n* = 5, 3.4%, *p* = 0.616), and dog bites (*n* = 1, 0.7%, *p* = 0.513).

Concerning all CMF and dental trauma patients, the propensity to violence correlated with the amount of alcohol consumed *(r* = 0.239, *p* = 0.001, *n* = 145), was associated with the incidence of facial fractures (*r* = 0.186, *p* = 0.031, *n* = 145), and a significant correlation with the male gender (*r* = 0.302, *p* = 0.001, *n* = 145). 

The distribution of facial fractures, soft tissue injuries and facial contusions, as well as other CMF concerns, are summarized in Figure 4. For each type of injury, more than one choice was possible.

### 3.5. Hospitalisation

Outpatient treatment could be carried out in 100% of dental patients (*n* = 53). All dental patients were referred for further treatment to the family dentist or dental clinic. Two patients were monitored in the ED overnight due to alcohol intoxication and for social reasons and sent to the dentist the following day. CMF patients were hospitalized and needed further treatment under general anesthesia in 20.9% of the cases.

### 3.6. Costs Distribution

The cost distribution between the two groups was as expected, namely that dental patients (mean value 710.36 Swiss francs, SD = 632.16) required lower costs than CMF patients (mean value 2948.92 Swiss francs, SD 5241). Patients who presented with toothache in ED generated the lowest costs. The most expensive conditions were fractures of the jaw and face, as well as infections that required the oral surgeon and in-patient treatment. No gender-specific differences were found. Annual costs were approximately equivalent for men and women.

Thirty-five (35) patients (*n* = 35, 17.4%) who presented to the ED solely for toothache gave rise to an average cost of 459 Swiss francs (SD 376.27). These patients were treated on an outpatient basis, usually by means of pain-relieving therapy and the recommendation to consult a private dentist. 

A total of 15 patients suffered from a dental infection, clinically manifested as swelling. Ten patients presented with a widespread infection, nevertheless limited to the alveolar ridge. This was treated locally and did not necessitate inpatient care. The costs were calculated as 1188.89 Swiss francs. 

The remaining five cases demonstrated advanced inflammation with poor general condition. This required surgical intervention under general anesthesia, as well as under inpatient conditions, resulting in mean costs of 6406 Swiss francs (range of 3115–10,660 Swiss francs) and an average hospitalization period of 3.6 days. In isolated cases, abscess relief was performed as an emergency. The average costs amounted to 618.82 Swiss francs (range of 74–2785 Swiss francs).

Dental trauma tended to be less expensive and gave a mean cost of 1612.60 Swiss francs (range of 208–6788 Swiss francs), as these patients were usually referred to dental colleagues. Tooth luxation injuries could sometimes be treated by the patient him/herself in an emergency setting and were therefore charged at a higher rate. The highest costs were compared in two cases where the patient had to be hospitalized and treated due to further facial fractures. In 13 cases, additional soft tissue injuries had to be treated.

The mean cost of facial fractures was estimated to be 5048 Swiss francs. The mean values of all annual costs were very heterogeneous.

## 4. Discussion

This study examines the characteristics and economic impact of dental and craniomaxillofacial conditions in immigrants from low- and middle-income countries. 

### 4.1. Oral Health of Immigrants

Dental visits to the ED have steadily increased over the last decade [23]. These patients are often unaware of other possible care settings [3,24,25,26]. The costs for dental treatment in Switzerland are usually borne by the patients themselves, which means that financially insecure patients might avoid visits for preventive purposes and treatments. These patients will seek help in an ED when suffering from an acute problem, as costs will then be covered by the compulsory health insurance. Especially immigrants from different cultural backgrounds are not familiar with preventive care services, such as routine screening, which can detect early oral health problems before they become symptomatic, expensive, and potentially damaging [27,28,29,30,31]. The key problem is that EDs are normally not equipped to provide definitive dental care. Especially in rural regions, EDs are not supported by a dental or maxillofacial department and are only able to relieve pain or infection by providing symptomatic-conservative treatment with analgesics or antibiotics—without curing the dental condition [32,33,34]. 

In the present study, no annual increase in dental visits in ED was observed. The number of visits to the ED with dental conditions only were 26.4% of all oral/craniomaxillofacial visits, i.e., less than 20 per year. Various economic and practical barriers may be mentioned as potential reasons, including the ignorance of other dental care options, the unwillingness to pay the costs or to wait for a dental appointment, language barriers, and that social isolation may influence decision-making [26,35,36]. It is not surprising that immigrants with social support, even if only in the form of family members and/or friends, are more informed about (oral) health care facilities [26,28]. Language barriers and poor education are crucial and may lead to misunderstandings. Those affected cannot obtain adequate information about where to seek help. These factors may result in the inappropriate usage of emergency services [25,29]. 

Most immigrants seeking help due to oral health problems were young (average age 35.69 years, SD 11.49). A study in a Swiss Urgent Care Centre calculated a similar average age (39.70 years) for Southeast European immigrants presenting at an ED, which was lower than the calculated mean age of the Swiss population (51.30 years) during the same period [3] The same study recorded more immigrants of lower triage level and fewer patients of higher triage level and more frequent self-reference than Swiss patients. Similarly, in our study, self-referencing was significantly higher compared to other referral types. Lower triage levels were likewise the most frequently recorded. 

The calculated average costs for dental problems in our ED was 459.51 Swiss francs per patient, with less than 20 consultation annually. In relation to the entire country, this could have some impact on the Swiss health care system but seems to be minor in relation to the overall costs.

### 4.2. Facial Injuries and Fractures 

Epidemiological data on facial fractures vary internationally and are greatly influenced by socioeconomic factors, specific time trends, and population, although, to our knowledge, no study on immigrants in Switzerland has yet been conducted [37].

Many maxillofacial injuries are not life-threatening, but may still lead to functional deficits with dramatic impact on the patient’s quality of life, and thus justify an ED visit [11,12]. The peak incidence of maxillofacial injury was observed in the age group of 16–25 years, with males outnumbering females in all age groups. Maximum numbers of trauma cases are reported in the late evening, independent of the weekday. These data correspond to those of other epidemiological studies [38].

The present study shows that facial fractures are associated with the male gender, alcohol consumption, and propensity to violence, which is in line with other findings. LMIC immigrants tend to be prone to violence due to individual risk factors such as cultural background, male gender, alcohol intoxication, substance abuse, and language barriers [39]. 

The second most common cause of facial injuries was falls, which is consistent with the literature. Due to demographic changes and the increasing aging of the population, the fall rate is continuously increasing. Falls constitute a major health concern, which can lead to numerous surgical interventions, prolonged hospitalization, and associated higher costs, not only due to CMF issues, but also because of orthopedic injuries. Several risk factors are linked to falls, contrasting intrinsic (muscle weakness of the lower limbs, poor grip strength, balance problems, functional, cognitive and visual impairment, dizziness, the use of assistive devices, depression, chronic illness, increasing age, prior history of falls, and female gender) to extrinsic factors (polypharmacy, drugs, low income, and environmental hazards, such as poor lighting, loose carpets, and lack of safety equipment) [40].

In contrast to studies from other countries, where traffic accidents are the most common cause of facial injuries, the numbers of road accidents in Switzerland are low. This could certainly be attributed to passive safety systems (airbag, early warning system for collisions, etc.) in modern vehicles, as well as the traffic rules and the mandatory use of seatbelts and helmets, as well as strict penalties for drunk driving [37].

In the present study, comparatively few dental traumas were reported. This may be because personnel without dental training have limited knowledge of tooth injuries, especially those in the posterior region, which are not clearly visible during the initial examination. Many patients in this study sustained only soft tissue injuries without bone involvement, so that some cases were not detected at the time of first presentation. This issue has also been described by other authors [41].

The epidemiological understanding of facial trauma and oral health in immigrants can help to avoid injuries, lower costs, and enhance the efficient management of public resources by emphasizing integration through public health education or the use of social networks and promoting appropriate preventive measures.

### 4.3. Limitations

Our study has some limitations. It is of a retrospective nature. Moreover, we included all immigrants in our study, regardless of generation. As a result, we may have overlooked important differences between the first and second generations, who are generally better integrated and have a more similar lifestyle to the native population than their parents. 

We only included data from an ED in central Switzerland, where the annual number of patients, including migrants, is higher than in other EDs of private hospitals in Bern or in the rest of the country. 

## 5. Conclusions

The etiology of dental and maxillofacial injuries in immigrants in Switzerland requires better support in the prevention of violence and continued promotion of oral health education.

## Figures and Tables

**Figure 1 ijerph-17-04906-f001:**
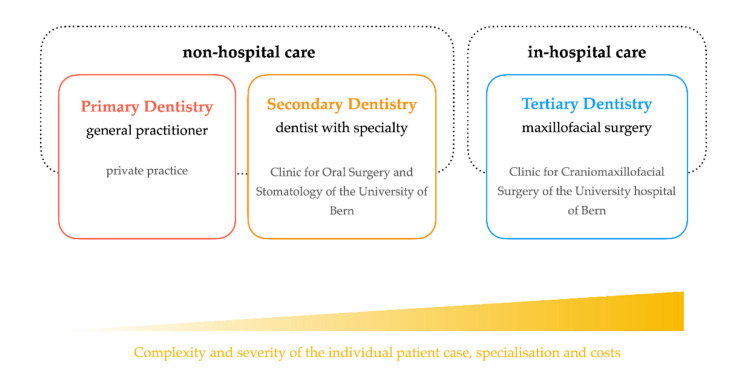
Illustration of the Bern model: Explanation of the terms “primary,” “secondary,” and “tertiary” dentistry.

**Figure 2 ijerph-17-04906-f002:**
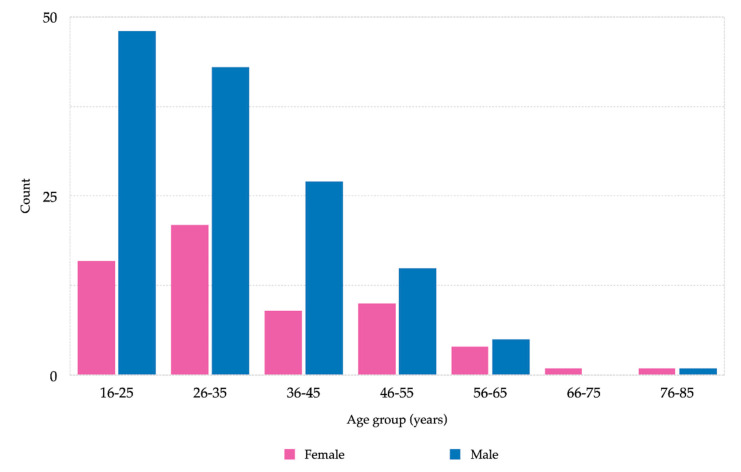
Total population by gender and age (*n* = 201).

**Figure 3 ijerph-17-04906-f003:**
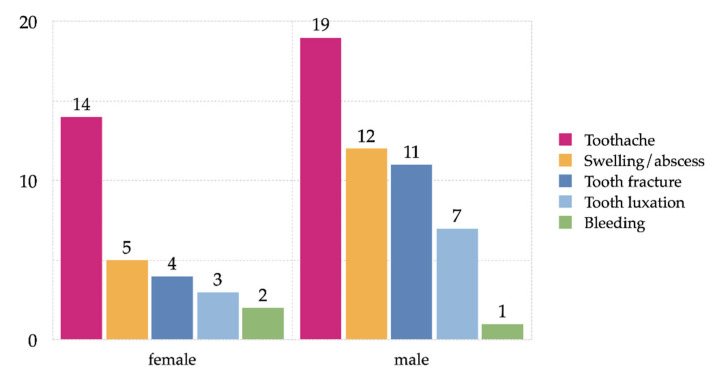
Overall summary of dental conditions by gender (*n* = 53).

**Figure 4 ijerph-17-04906-f004:**
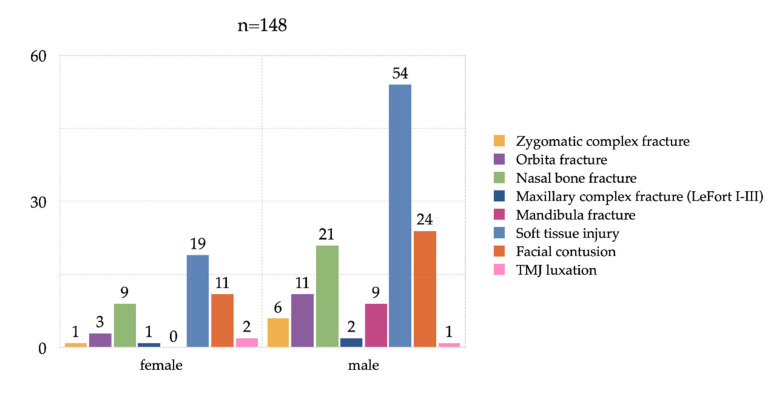
Distribution of CMF emergencies by gender (*n* = 148).

**Table 1 ijerph-17-04906-t001:** Distribution of age groups, gender, and nationality between dental patients (Dental) and craniomaxillofacial (CMF) emergencies. Legend: ^a^ Total percentage within the columns; ^b^ Total percentage within the rows; CMF: Craniomaxillofacial.

Baseline Characteristics		Dental			CMF			Total			*p*-Value
		*n*	% ^a^	% ^b^	*n*	%	% ^b^	*n*	%	% ^b^	
**Age group (years)**	16–25	9	17.0	14.1	55	37.2	85.9	64	31.8	100	0.009
	26–35	20	37.7	31.3	44	29.7	68.7	64	31.8	100	
	36–45	12	22.6	33.3	24	16.2	66.7	36	17.9	100	
	46–55	12	22.6	48.0	13	8.8	52.0	25	12.4	100	
	56–65	0	0.0	0.0	9	6.1	100	9	4.5	100	
	66–75	0	0.0	0.0	1	0.7	100	1	0.5	100	
	76–85	0	0.0	0.0	2	1.4	100	2	1.0	100	
**Gender**	male	35	66.0	25.2	104	70.3	74.8	139	69.2	100	0.567
	female	18	34.0	29.0	44	29.7	71.0	62	30.8	100	
**Nationality group**	low income	7	13.2	25.0	21	14.2	75.0	28	13.9	100	0.363
	low middle income	24	45.3	32.0	51	34.5	68.0	75	37.3	100	
	upper middle income	22	41.5	22.4	76	51.4	77.6	98	48.8	100	
	Total	53	100	26.4	148	100	73.6	201	100	100	

**Table 2 ijerph-17-04906-t002:** Emergency room related data categorized according to dental (Dental) and craniomaxillofacial (CMF) emergency consultations.

Emergency Room Variables	Subcategories	Dental			CMF			Total			*p*-Value
		*n*	% ^a^	% ^b^	*n*	%	% ^b^	*n*	%	% ^b^	
Type of referral	Self-referral	46	86.8	35.9	82	55.4	64.1	128	63.7	100	0.001
	Ambulance	3	5.7	7.1	39	26.4	92.9	42	20.9	100	0.001
	External hospital	0	0.0	0.0	11	7.4	100	11	5.5	100	0.041
	Police/prison	1	1.8	11.1	8	5.4	88.9	9	4.5	100	0.288
	General practitioner	2	3.8	28.6	5	3.4	71.4	7	5.5	100	0.893
	Internal hospital	1	1.9	33.3	2	1.4	66.7	3	1.5	100	0.783
	Swiss Air rescue	0	0.0	0.0	1	0.7	100.0	1	0.5	100	0.549
Time of admission	2016	16	30.2	28.6	40	27.0	71.4	56	27.9	100	0.892
	2017	16	30.5	25.0	48	32.4	75.0	64	31.8	100	
	2018	12	22.6	23.5	39	26.4	76.5	51	25.4	100	
	2019	9	17.0	30.0	21	14.2	70.0	30	14.9	100	
	weekdays (Mon-Fri)	29	55.0	33.3	57	44.9	57.6	99	49.3	100	0.354
	weekends (Sat-Sun)	24	45.0	26.5	70	55.1	68.6	102	50.7	100	
	06:01–12:00	6	13.3	25.0	18	12.2	75.0	24	11.9	100	0.947
	12:01–18:00	17	32.1	27.9	44	29.7	72.1	61	30.3	100	
	18:01–06:00	30	56.6	25.9	86	58.1	74.1	116	57.7	100	
Triage	1	0	0.0	0.0	0	0.0	0.0	0	0	100	0.001
	2	0	0.0	0.0	20	13.5	100.0	20	10	100	
	3	42	79.2	24.9	127	85.8	75.1	169	84.1	100	
	4	11	20.8	91.7	1	0.7	8.3	12	6.0	100	
	5	0	0.0	0.0	0	0.0	0.0	0	0	100	
Reason for admission	Toothache	33	62.3	100	0	0.0	0.0	33	16.4	100	0.001
	Swelling	11	20.8	64.7	6	4.1	35.3	17	8.5	100	0.001
	Lockjaw	0	0.0	0.0	3	2.0	100	3	1.5	100	0.296
	Bleeding	1	1.9	33.3	2	1.4	66.7	3	1.5	100	0.783
	Trauma	8	15.1	5.5	137	92.6	94.5	145	72.1	100	0.001
Primary Survey	Staff ED (Surgery)	38	71.7	25.9	109	73.6	74.1	147	73.1	100	0.926
	Staff ED (FTN)	11	20.8	26.8	30	20.3	73.2	41	20.4	100	
	CMF Surgeon	4	7.5	30.8	9	6.1	69.2	13	6.5	100	
Sec. Survey	CMF Surgeon	9	17.0	12.9	61	41.2	87.1	70	34.8	100	0.001
Hospital stays	Out-patient	53	100	31.2	117	79.1	68.8	170	84.6	100	0.001
	In-patient	0	0.0	0.0	31	20.9	100	31	15.4	100	
	Total	53	100	26.4	141	100	73.6	201	100	100	

Legend: ^a^ Total percentage within the columns; ^b^ Total percentage within the rows; FTN Fast Track/GP.

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
