# Peer review of "Violence as the Most Frequent Cause of Oral and Maxillofacial Injuries among the Patients from Low- and Middle-Income Countries—A Retrospective Study at a Level I Trauma University Emergency Department in Switzerland"

_ijerph, 2020, doi:10.3390/ijerph17134906_

Round 1

Reviewer 1 Report

IJERPH – 833582

This is a retrospective analysis of administrative data from an emergency department (ED) located at a University hospital in Bern, Switzerland during the period 2016 - 2019. The authors provide descriptive analyses of the types of dental and trauma cases presenting at the ED associated with patients from specific nationality groups defined as being from a low-middle income country.

This analysis is important contribution for helping to improve understanding of the health experiences of immigrants and improving the health care system’s response to managing their emergency needs; and also for identifying appropriate and effective prevention strategies at policy levels.

Before considering this paper to be ready for publication I think the authors need to address some important issues to clarify their methods, results and interpretation of the data.

  1. Study design – the authors need to provide details of the study period - that is the months/years that data were obtained.
  2. Line 88 – the closing bracket for age groups is in the wrong place and needs to be corrected.
  3. Nationality groups – the authors need to define how they classified these groups. Did they use OECD definitions? World Bank definitions? Other?
  4. Line 89 – The authors need to define what the triage levels are and what they mean. Perhaps this could be reported in a Table for ease of reading. I see they refer to Manchester Triage System in Section 2.4, but how this still needs to be defined for the reader, especially as they are seeking publication in a general public health journal not a oral/dental traumatology journal.
  5. The Bern Model – I see that there are 3 levels of provision of dentistry in Bern but do not see how this is relevant for the analyses conducted in this study. I think description/contextualisation of this belongs in the Introduction rather than in the Methods. I assume from this that all cases analysed in this study were secondary or tertiary level but this is not reported in the Results that I can see. How does this model link with attendance at ED?
  6. Statistical analyses: the authors report that linear regression analyses were conducted but no details are provided – for which analyses? Were confounding variables controlled for? This section requires greater detail.
  7. Results Line 125 – the authors report the sample as 201, but in the Methods it is 202 – which is correct?
  8. Results: Figure 2 is not the correct way to visualise the demographic distribution of the sample - this should be done with a column graph as these relationships are not time trends. The lay out of Table 1 is distorted – this needs to be corrected, especially for column “Dental % “.
  9. Results Table 2: I think ‘Art of referral” should be “Type of referral”. There is an extra line in the CMF columns associated with Assault which makes reading the Table very difficult. I suggest placing the footnote about the domestic violence figures as an actual footnote to the Table rather than within the Category – it is confusing.
  10. Results, 3.5 Hospitalisation: The authors report that outpatient treatment could be carried out in over ¾ of dental patients but the figure in the brackets is 96.2% - I suggest the authors provide the more accurate result.
  11. Results, Cost distribution: The Table reported here is in a very different format to other Tables. I assume the journal may request the table to be consistent with the others. However, I am curios as to why the authors are reporting standard deviation, range (maximum and minimum) and then also report 95% confidence interval. What is the purpose of 95%CI in this context?
  12. Discussion: Line 267 – I think the word ‘care’ is missing after ‘preventive’.
  13. Line 291: The authors state that this study backs up/confirms that LMIC immigrants tend to be prone to violence – I do not think that this study, which reports only descriptive analyses of anonymised administrative data, can make this statement or claim. On what basis does this study address this question? There are complex sociocultural factors that contribute to disadvantage which may be a confounding factor underpinning the presentations of these residents to ED for treatment. I suggest the authors reconsider what claim they are making here and provide evidence to back up this judgement.
  14. Line 298: The authors need to provide some detail on the risk factors linked to falls as I am not sure what they mean here.
  15. Line 306 & 307: I think the word ‘scanty’ should be replaced with ‘limited’. Also ‘first’ should be replaced with ‘initial’.
  16. Line 310: In what way does awareness of characteristics of facial trauma help avoid injuries? The injury has already occurred by the time it is assessed. Do the authors mean that early identification and early treatment of all levels of injuries reduces complications? Or something else?
  17. Line 311: The authors state “enhance management of public resources by emphasising adequate preventive measures” – this is unclear. I am not sure what the authors mean by this – can they be more specific?
  18. Line 315: The authors refer to “generation” - Do the authors mean generations that have been resident in Switzerland? How might this be factor in their presentation to ED? Or in fact to have a dental infection or experience trauma? Does ‘integration’ reduce risk?
  19. Why is limiting the data to adult patients a limitation? Surely analysis of patterns of dental problems and trauma in children would be separate anyway? Their demographics and risk profiles differ greatly from adolescents, young adults, middle age and older adults.
  20. Conclusions: I am not comfortable with the authors drawing a conclusion from this study that includes “strict legislation is required against violence” - on what basis do they conclude this? What about addressing the underlying sociocultural aspects that underpin violence? Is this a health/human right issue or just a legal issue? What about falls prevention? What about reducing the barriers to accessing the general dental practitioner for dental problems? How about improved understanding of the drivers that underlie the decision that LMIC immigrants make to attend an ED for a health issue?

Author Response

Bern, June 15th

Dear Reviewers,

On behalf of my co-authors, I would like to kindly thank you for your review of our manuscript, entitled “Violence as the most frequent cause of oral and maxillofacial injuries among the patients from low and middle income countries – a retrospective study at a level I Trauma University Emergency Department in Switzerland”. I would also like to thank you for giving us the opportunity to re-submit our article and for your very helpful comments.

As such, the authors have carefully assessed the Reviewer’s comments and the responses to the Reviewer’s comments are as follows as an answer to the each of the following addressed comments:

Reviewer #1:

  1. “Study design – the authors need to provide details of the study period – that is the months/years that data were obtained.”
  • Author Response: We have added this information in M&M and Results section.

  1. “Line 88 – the closing bracket for age groups is in the wrong place and needs to be corrected.”

  • Author Response: Thank you for pointing that out. We have adjusted this accordingly.

  1. “Nationality groups – the authors need to define how they classified these groups. Did they use OECD definitions? World Bank definitions? Other?”
  • Author Response: Classification of the countries was based on income according to World Bank Country and Lending Groups: Low income countries ($1,025 or less), Low middle income countries ($1,026 to $3,995), Upper middle income countries ($3,996 to $12,375). We have added the relevant information in the Material & Methods section.

  1. “Line 89 – the authors need to define what the triage levels are and what they mean. Perhaps this could be reported in a Table for ease of reading. I see they refer to Manchester Triage System in Section 2.4, but how this still needs to be defined for the reader, especially as they are seeking publication in a general public health journal not a oral/dental traumatology journal.”
  • Author Response: Thank you for the comment. The information has been added in the text. This triage system classifies the urgency of a treatment for patients in five different priority levels: 1-acute life threating problem (immediate treatment required), 2-high urgency, 3-urgency, 4-less urgency, 5-non urgency.”

  1. “The Bern Model – I see that there are 3 levels of provision of dentistry in Bern but do not see how this is relevant for the analyses conducted in this study. I think description/contextualisation of this elongs in the Introduction rather than in the Methods. I assume from this that all cases analysed in this study were secondary or tertiary level but this is not reported in the Results that I can see. How does this model link with attendance at ED?”

  • Author Response: Thank you very much for this advice. We have tried to explain the context of the Bern Model in the introduction. The Bern Model actually only refers to oral health and not to facial fractures. For oral health, the 3 levels are frequently not considered and the patients choose the easiest way by going directly to the emergency department. We wanted to show that many oral health referrals should/could have been treated primarily in a private practice or dental clinic rather than in the ED/maxillofacial surgery. We have mentioned this in more detail in the results part and in the discussion.

  1. “Statistical analyses: the authors report that linear regression analyses were conducted but no details are provided – for which analyses? Were confounding variables controlled for? This section requires greater detail.”

  • Author Response: Thanks for this advice. The "statistical analysis" section has been adjusted accordingly. The corresponding P-values in the respective tables were added using Chi-Square test. Correlations were made according to Pearson. In particular, correlations between the occurrence of facial fracture/trauma with violence, alcohol consumption and gender. We've highlighted this in a better way.

  1. “Results Line 125 – the authors report the sample as 201, but in the Methods, it is 202 – which is correct?”

  • Author Response: Thank you, this was a typo. We've corrected the error.

  1. “Results: Figure 2 is not the correct way to visualize the demographic distribution of the sample - this should be done with a column graph as these relationships are not time trends. The layout of Table 1 is distorted – this needs to be corrected, especially for column “Dental % “.”

  • Author Response: Figure 2 has been changed to a bar chart accordingly. We adjusted the lay out of Table 1.

  1. “Results Table 2: I think ‘Art of referral” should be “Type of referral”. There is an extra line in the CMF columns associated with Assault which makes reading the Table very difficult. I suggest placing the footnote about the domestic violence figures as an actual footnote to the Table rather than within the Category – it is confusing.”

  • Author Response: We have adjusted Table 2. "Art of referral" has been renamed "Type of referral." They have also been recalculated and renamed.

  1. “Results, 3.5 Hospitalisation: The authors report that outpatient treatment could be carried out in over ¾ of dental patients but the figure in the brackets is 96.2% - I suggest the authors provide the more accurate result.”

  • Author Response: We totally agree and have changed that.

  1. “Results, Cost distribution: The Table reported here is in a very different format to other Tables. I assume the journal may request the table to be consistent with the others. However, I am curious as to why the authors are reporting standard deviation, range (maximum and minimum) and then also report 95% confidence interval. What is the purpose of 95%CI in this context?”
  • Author Response: We totally agree. We have deleted table 3.

  1. “Discussion: Line 267 – I think the word ‘care’ is missing after ‘preventive’”

  • Author Response: Thank you for the advice. We corrected that.

  1. “Line 291: The authors state that this study backs up/confirms that LMIC immigrants tend to be prone to violence – I do not think that this study, which reports only descriptive analyses of anonymized administrative data, can make this statement or claim. On what basis does this study address this question? There are complex sociocultural factors that contribute to disadvantage which may be a confounding factor underpinning the presentations of these residents to ED for treatment. I suggest the authors reconsider what claim they are making here and provide evidence to back up this judgement.”

  • Author Response: We absolutely agree. We left the quote alone.

  1. “Line 298: The authors need to provide some detail on the risk factors linked to falls as I am not sure what they mean here.”

  • Author Response: The sentence was incomplete and confusing. We have redrawn the meaning and added examples for “intrinsic and extrinsic factors” for falls.

  1. “Line 306 & 307: I think the word ‘scanty’ should be replaced with ‘limited’. Also ‘first’ should be replaced with ‘initial’.”

  • Author Response: We have replaced the two words.

  1. “Line 310: In what way does awareness of characteristics of facial trauma help avoid injuries? The injury has already occurred by the time it is assessed. Do the authors mean that early identification and early treatment of all levels of injuries reduces complications? Or something else?”

  • Author Response: What we actually meant here is, that fractures might be avoided through appropriate integration, preventive measures, information and education. In addition, oral health should be trained through oral hygiene instructions, smoking prevention, etc. We have rewritten this accordingly, so that the meaning of the sentence is better understood.

  1. “Line 311: The authors state “enhance management of public resources by emphasizing adequate preventive measures” – this is unclear. I am not sure what the authors mean by this – can they be more specific?”

  • Author Response: We have improved this in the context of answer 16.

  1. “Line 315: The authors refer to “generation” - Do the authors mean generations that have been resident in Switzerland? How might this be factor in their presentation to ED? Or in fact to have a dental infection or experience trauma? Does ‘integration’ reduce risk?”

  • Author Response: By “generation” we mean those who were born in Switzerland but do not have a Swiss passport. We think that integration does reduce risk and may have an impact on the presentation in the ED or on dental infections/trauma. In Switzerland the prevention and education of oral health is beginning in primary school. Since we have not studied children, it can be assumed that adults, who have attended school in Switzerland, should have a better understanding of oral health.

  1. “Why is limiting the data to adult patients a limitation? Surely analysis of patterns of dental problems and trauma in children would be separate anyway? Their demographics and risk profiles differ greatly from adolescents, young adults, middle age and older adults.”

  • Author Response: We totally agree and have removed this sentence.

  1. Conclusions: I am not comfortable with the authors drawing a conclusion from this study that includes “strict legislation is required against violence” – on what basis do they conclude this? What about addressing the underlying sociocultural aspects that underpin violence? Is this a health/human right issue or just a legal issue? What about falls prevention? What about reducing the barriers to accessing the general dental practitioner for dental problems? How about improved understanding of the drivers that underlie the decision that LMIC immigrants make to attend an ED for a health issue?”

  • Author Response: We totally agree. The statements were imprecise. We have rewritten the conclusion.

I would also like express our gratitude to the editors and reviewers for their careful and diligent review. If you need further information, please do not hesitate to contact me.

Best regards,

John Patrik Burkhard

Department of Cranio-Maxillofacial Surgery Inselspital

Bern University Hospital

University of Bern

Switzerland

Electronic address: jp.burkhard@insel.ch

On behalf of all co-authors.

Corresponding author: John Patrik Burkhard, Department of Cranio-Maxillofacial Surgery Inselspital. Bern University Hospital, University of Bern, Switzerland. Electronic address: jp.burkhard@insel.ch

Reviewer 2 Report

General Comments

  • You present lots of data, and lots of graphs, and it gets very confusing and hard to follow! I wonder if there is an easier way to present this data or at least reduce the number of graphs currently in use!
  • Please don’t use pie charts. It isn’t scientific and looks amateurish.
  • The results section of the paper is underwhelming as all you have done is present descriptive statistics, and not looked for any correlations or associations. Or a regression analysis as promised.
  • Your discussion section, in parts, doesn’t actually discuss the result of your study. It is more like a literature review.
  • It seems like immigrants attending ED is portrayed to be a significant problem. However,  if approx. 30% of the population are immigrants, and most will live in cities or the capital, then to only have 201 attend over 3.5 year period is quite small? Would the cost of setting up early diagnosis for this group not cost more? However, it is unlikely to reduce inequalities that these groups face?  You contradict yourself as in your limitation section you mention “ We only included data from an ED in central Switzerland, where the annual number of patients, including migrants, is higher than in other EDs of private hospitals in Bern or in the rest of the country”  - therefore, is approx. 60 patients per year such a big problem? How does this compare to attendances for non-immigrants?
  • There is a lack of generalisability of your study to other European countries? Is this a major issue in other countries?  This hasn’t been discussed.

Specific comments:

  • I like Table 1. Really nice way to present that data, and it is the only clear table you present.
  • It would be nice to have a description of what low, low middle and upper middle-income boundaries were? Is there an upper?
  • Section 2.4 – need to explain what each level is, especially if you go into that detail in the results.
  • During statistical analysis, was a test of normality done to ensure data was analysed appropriately? I.e. age isn't always normally distributed 
  • Line 126 - "Over a period of four years from January 2016 until September 2019” - this is not a 4 year period, please amend
  • 201 patients with migration background were admitted to the ED with oral or craniomaxillofacial (CMF) problems - I am assuming this is the number of patients that were seen by a dentist or craniofacial team, or the total number who attended the ED, with some being turned away after nurse triaging?   This needs to be clearer.
  • Were the records all written by a dental or CMF member of staff? This needs to be clearer.
  • Was there any missing data? It seems unusual that you have all the data for all 201 patients.
  • In table 2, I would split up Assault and DV into two rows (incl. domestic violence, n=10, 6.8%)
  • In table 2, I would put lay out art of referral and reason for admission in ascending order.
  • I would argue that “bicycle” and “sport” are not reasons for referral, whereas “dental caries “ is. The former are mechanisms of injury.  I would say you need to address this and make it clearer that the reasons for referral were:
    • Dental trauma, Toothache, Maxillofacial Trauma, Swelling etc.
  • In table 2, I am also surprised there is no “Swelling” as a cause for admission. If this is included in Carious Teeth, then I would split up.
  • In table 2, if the reason for admission was only relevant to CMF, then you need to ensure you put 0 in the dental column to make the table flow. At present by omitting it looks messy.
  • In table 2, time of admission, do you have details for Saturday and Sunday, if they differ I would be inclined to include. Also, by starting at 6am on a Saturday morning, does that mean anything from 12 midnight – 6am Saturday morning isn’t the weekend? Maybe need to change the timings if possible.
  • In table 2, Triage (according to the Swiss system) but in the manuscript you say Manchester system? Which did you use?
  • Section 3.4 is just repetition of the Table 2. This needs to be presented in one, not both, ways. Please see comment above re: infection etc
  • Line 193 - “three quarters” rather than “3/4”
  • Table 3 is not needed!
  • You say “Categorical variables were analysed using the chi‐square (χ2 ) test and the Fisher exact test. Linear regression analyses were performed” – I see one P value and no liner regression.  Your analysis is just descriptive!
  • How were your costs calculated? There is not mention of this, or accounting for inflation/variation over the 3.5 year period.
  • Line 253 – A range is needed here as this seems relatively small in comparison to other costs presented. As you have a number, provide a total costs of all treatments for this group. 
  • Line 289-290 – why are p-values presented in the discussion but not the results?

Author Response

Bern, June 15th

Dear Reviewers,

On behalf of my co-authors, I would like to kindly thank you for your review of our manuscript, entitled “Violence as the most frequent cause of oral and maxillofacial injuries among the patients from low and middle income countries – a retrospective study at a level I Trauma University Emergency Department in Switzerland”. I would also like to thank you for giving us the opportunity to re-submit our article and for your very helpful comments.

As such, the authors have carefully assessed the Reviewer’s comments and the responses to the Reviewer’s comments are as follows as an answer to the each of the following addressed comments:

Reviewer #2:

  1. “You present lots of data, and lots of graphs, and it gets very confusing and hard to follow! I wonder if there is an easier way to present this data or at least reduce the number of graphs currently in use!
  • Author Response: We have reduced and revised the graphics and tables to show the data more clearly.

  1. Please don’t use pie charts. It isn’t scientific and looks amateurish.
  • Author Response: Thanks for pointing that out. We've abandoned pie charts.

  1. The results section of the paper is underwhelming as all you have done is present descriptive statistics, and not looked for any correlations or associations. Or a regression analysis as promised. Your discussion section, in parts, doesn’t actually discuss the result of your study. It is more like a literature review.
  • Author Response: We have added the missing statistical values to the results and adapted the discussion, removed irrelevant passages and tried to better discuss the results with the current literature.

  1. It seems like immigrants attending ED is portrayed to be a significant problem. However, if approx. 30% of the population are immigrants, and most will live in cities or the capital, then to only have 201 attends over 3.5-year period is quite small? Would the cost of setting up early diagnosis for this group not cost more? However, it is unlikely to reduce inequalities that these groups face? You contradict yourself as in your limitation section you mention “We only included data from an ED in central Switzerland, where the annual number of patients, including migrants, is higher than in other EDs of private hospitals in Bern or in the rest of the country” - therefore, is approx. 60 patients per year such a big problem? How does this compare to attendances for non-immigrants? There is a lack of generalizability of your study to other European countries? Is this a major issue in other countries? This hasn’t been discussed.”
  • Author Response: You're absolutely right. In our study, we may have overemphasized the problem. However, we found it quite important to identify and analyze the problem early on. We wanted to avoid stigmatizing a population group in order to improve integration and to increase immigrants' knowledge of the resources of our health system. Actually, we wanted to find out what the circumstances really were. Since there are no studies on this subject in Switzerland. There are many regional hospitals in Switzerland, especially in the city of Bern, so the cases are certainly distributed. However, according to our experience and our contacts with other hospitals, they have less frequent patients with oral/maxillofacial injuries. Of course, a comparison with the local population would have been more meaningful. We have adapted the wording and discussed the results accordingly.

  1. “I like Table 1. Really nice way to present that data, and it is the only clear table you present.”
  • Author Response: Thank you for the advice. We have tried to present the other tables and graphics more clearly.
  1. “It would be nice to have a description of what low, low middle and upper middle-income boundaries were? Is there an upper?”
  • Author Response: Classification of the countries was based on income according to World Bank Country and Lending Groups: Low income countries ($1,025 or less), Low middle income countries ($1,026 to $3,995), Upper middle income countries ($3,996 to $12,375). We have added the relevant information in the Material & Methods section.

  1. “Section 2.4 – need to explain what each level is, especially if you go into that detail in the results.”
  • Author Response: Thank you for the comment. The information has been added in the text. This triage system classifies the urgency of a treatment for patients in five different priority levels: 1-acute life threating problem (immediate treatment required), 2-high urgency, 3-urgency, 4-less urgency, 5-non urgency.”

  1. “During statistical analysis, was a test of normality done to ensure data was analyzed appropriately? I.e. age isn't always normally distributed.”
  • Author Response: Yes, the normal distribution of the data was tested.

  1. “Line 126 - "Over a period of four years from January 2016 until September 2019” - this is not a 4-year period, please amend”

  • Author Response: Thanks for that. We changed it.

  1. “201 patients with migration background were admitted to the ED with oral or craniomaxillofacial (CMF) problems - I am assuming this is the number of patients that were seen by a dentist or craniofacial team, or the total number who attended the ED, with some being turned away after nurse triaging? This needs to be clearer.”

  • Author Response: All patients were examined by a physician. Milder cases were triaged in a family doctor's practice integrated into the ED. We have described this more clearly.

  1. “Were the records all written by a dental or CMF member of staff? This needs to be clearer.”

  • Author Response: We have added this in M&M section.

  1. “Was there any missing data? It seems unusual that you have all the data for all 201 patients.”

  • Author Response: for our study we have not observed any missing data. In our patient system (E.care) the medical history, diagnosis, treatment procedure, etc. are recorded very precisely. For patients who were hospitalized, we also used the treatment, surgery and discharge reports from the wards for data collection.

  1. “In table 2, I would split up Assault and DV into two rows (incl. domestic violence, n=10, 6.8%)”

  • Author Response: We have mentioned the DV as a footnote.

  1. “In table 2, I would put lay out art of referral and reason for admission in ascending order.”

  • Author Response: We have adjusted this accordingly.

  1. “I would argue that “bicycle” and “sport” are not reasons for referral, whereas “dental caries“ is. The former are mechanisms of injury. I would say you need to address this and make it clearer that the reasons for referral were: Dental trauma, Toothache, Maxillofacial Trauma, Swelling etc.?”
  • Author Response: We totally agree. We have captured the trauma mechanisms in the text and removed the values in the table to avoid duplication (see comment 20).

  1. “In table 2, I am also surprised there is no “Swelling” as a cause for admission. If this is included in Carious Teeth, then I would split up.”

  • Author Response: The swelling was always associated with bad teeth.

  1. “In table 2, if the reason for admission was only relevant to CMF, then you need to ensure you put 0 in the dental column to make the table flow. At present by mitting it looks messy.”

  • Author Response: Right, that is a bit confusing. we redefined it accordingly.

  1. “In table 2, time of admission, do you have details for Saturday and Sunday, if they differ, I would be inclined to include. Also, by starting at 6am on a Saturday morning, does that mean anything from 12 midnight – 6am Saturday morning isn’t the weekend? Maybe need to change the timings if possible”

  • Author Response: Weekends were calculated from 00:00 for 24 hours (i.e. the whole day) regardless of the time units listed below. The time span below should represent or define morning - noon/afternoon - evening/night.

  1. “In table 2, Triage (according to the Swiss system) but in the manuscript you say Manchester system? Which did you use?”

  • Author Response: Patients were routinely triaged in the Inselspital by using the Swiss Emergency Triage Scale. The latter is an abbreviated version of the validated Manchester Triage System (Mackway-Jones, K. Emergency Triage: Manchester Triage Group; BMJ Publishing Group: London, UK, 1997). This triage system stratifies the urgency of treatment for patients presenting to an ED by the five following levels: 1: Acute life threating problem (immediate treatment required); 2: High urgency; 3: Urgency; 4: Less urgency; and 5: No urgency. We have added appropriate information to the text to provide clarity.

  1. “Section 3.4 is just repetition of the Table 2. This needs to be presented in one, not both, ways. Please see comment above re: infection etc.”

  • Author Response: We have left this section as text and removed it from the table.

  1. “Line 193 - “three quarters” rather than “3/4”.”

  • Author Response: We have replaced 3/4 by the exact number.

  1. “Table 3 is not needed!”

  • Author Response: We have deleted table 3.

  1. “You say “Categorical variables were analysed using the chisquare (χ2) test and the Fisher exact test. Linear regression analyses were performed” – I see one P value and no liner regression. Your analysis is just descriptive!”

  • Author Response: We've rewritten the statistics and have indicated the tests performed accordingly.

  1. “How were your costs calculated? There is not mention of this, or accounting for inflation/variation over the 3.5 year period.”

  • Author Response: In Switzerland, there are two different methods for evaluating medical costs: (1) the Swiss medical currency is "tax points" (TP), or (2) the total cost of a patient. In our hospital, one TP corresponds to approximately 0.86 Swiss francs (about 0.87 US dollars). In our case, we have decided to compare the total costs of a patient without taking into account the cost difference associated with any inflation. Inflation was very low in Switzerland during this study period (2016 -0.43%; 2017 +0.54%; 2019 + 0.94%). In our view, there is no significant impact on the analyses and the results of the analyses.

  1. “Line 253 – A range is needed here as this seems relatively small in comparison to other costs presented. As you have a number, provide a total costs of all treatments for this group.”

  • Author Response: We have rewritten the discussion and listed the costs for those patients who were treated as outpatients with dental problems.

  1. “Line 289-290 – why are p-values presented in the discussion but not the results?”

  • Author Response: We have added the p-values in the results section.

I would also like express our gratitude to the editors and reviewers for their careful and diligent review. If you need further information, please do not hesitate to contact me.

Best regards,

John Patrik Burkhard

Department of Cranio-Maxillofacial Surgery Inselspital

Bern University Hospital

University of Bern

Switzerland

Electronic address: jp.burkhard@insel.ch

On behalf of all co-authors.

Corresponding author: John Patrik Burkhard, Department of Cranio-Maxillofacial Surgery Inselspital. Bern University Hospital, University of Bern, Switzerland. Electronic address: jp.burkhard@insel.ch

Round 2

Reviewer 1 Report

The authors have addressed all concerns I raised in the first review. The paper is now clearer and does not overstate it’s findings. 

Author Response

Bern, June 25th

Dear Reviewers,

On behalf of my co-authors, I would like to kindly thank you for your review of our manuscript, entitled “Violence as the most frequent cause of oral and maxillofacial injuries among the patients from low and middle income countries – a retrospective study at a level I Trauma University Emergency Department in Switzerland”. I would also like to thank you for giving us the opportunity to re-submit our article and for your very helpful comments.

As such, the authors have carefully assessed the Reviewer’s comments and the responses to the Reviewer’s comments are as follows as an answer to the each of the following addressed comments:

Reviewer #1:

  1. “The authors have addressed all concerns I raised in the first review. The paper is now clearer and does not overstate it’s findings.”
  • Author Response: Thank you for your review. We corrected grammar and spelling..

I would also like express our gratitude to the editors and reviewers for their careful and diligent review. If you need further information, please do not hesitate to contact me.

Best regards,

John Patrik Burkhard

Department of Cranio-Maxillofacial Surgery Inselspital

Bern University Hospital

University of Bern

Switzerland

Electronic address: jp.burkhard@insel.ch

On behalf of all co-authors.

Corresponding author: John Patrik Burkhard, Department of Cranio-Maxillofacial Surgery Inselspital. Bern University Hospital, University of Bern, Switzerland. Electronic address: jp.burkhard@insel.ch

Reviewer 2 Report

Thank you for addressing my comments. 

The manuscript now flows better and the tables are far clearer.   There are one or two spelling and grammatical errors.

Author Response

Bern, June 25th

Dear Reviewers,

On behalf of my co-authors, I would like to kindly thank you for your review of our manuscript, entitled “Violence as the most frequent cause of oral and maxillofacial injuries among the patients from low and middle income countries – a retrospective study at a level I Trauma University Emergency Department in Switzerland”. I would also like to thank you for giving us the opportunity to re-submit our article and for your very helpful comments.

As such, the authors have carefully assessed the Reviewer’s comments and the responses to the Reviewer’s comments are as follows as an answer to the each of the following addressed comments:

Reviewer #2:

  1. “The manuscript now flows better and the tables are far clearer. There are one or two spelling and grammatical errors.”
  • Author Response: Thank you for your review. We corrected grammar and spelling.

I would also like express our gratitude to the editors and reviewers for their careful and diligent review. If you need further information, please do not hesitate to contact me.

Best regards,

John Patrik Burkhard

Department of Cranio-Maxillofacial Surgery Inselspital

Bern University Hospital

University of Bern

Switzerland

Electronic address: jp.burkhard@insel.ch

On behalf of all co-authors.

Corresponding author: John Patrik Burkhard, Department of Cranio-Maxillofacial Surgery Inselspital. Bern University Hospital, University of Bern, Switzerland. Electronic address: jp.burkhard@insel.ch
